# Structures and Properties of Dinitrosyl Iron and Cobalt Complexes Ligated by Bis(3,5-diisopropyl-1-pyrazolyl)methane

**Haruka Kurihara, Ayuri Ohta and Kiyoshi Fujisawa *** 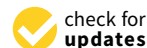

Department of Chemistry, Ibaraki University, Mito, Ibaraki 310-8512, Japan; haruka.k.crx0415@gmail.com (H.K.); 18nm005t@vc.ibaraki.ac.jp (A.O.)
* Correspondence: kiyoshi.fujisawa.sci@vc.ibaraki.ac.jp; Tel.: +81-29-853-8373

**Abstract:** Two dinitrosyl iron and cobalt complexes [Fe(NO)$_2$(L1″)](BF$_4$) and [Co(NO)$_2$(L1″)](BF$_4$) are synthesized and characterized, supported by a less hindered bidentate nitrogen ligand bis(3,5-diisopropyl-1-pyrazolyl)methane (denoted as L1″), are surprisingly stable under argon atmosphere. X-ray structural analysis shows a distorted tetrahedral geometry. Spectroscopic and structural parameters of the dinitrosyl iron and cobalt complexes are consistent with the previous reported {Fe(NO)$_2$}$^9$ and {Co(NO)$_2$}$^{10}$. Two N–O and M–N(O) stretching frequencies and their magnetic properties are also consistent with the above electronic structural assignments. We explored the dioxygen reactivities of the obtained dinitrosyl complexes. Moreover, the related [FeCl$_2$(L1″)], [Co(NO$_3$)$_2$(L1″)], and [Co(NO$_2$)$_2$(L1″)] complexes are also characterized in detail.

**Keywords:** nitric oxide; iron complex; cobalt complex; tetrahedral geometry; infrared spectroscopy

## 1. Introduction

Nitric oxide (NO) plays important roles in biology, such as nerve signal transduction, vasodilation, and immune defense [1]. The biosynthesis of NO and its physiological reactions reveals some interaction with metalloproteins [2–7]. NO has also been implicated in the disassembly of iron–sulfur proteins, leading to the formation of dinitrosyl iron complexes (DNICs) [8–10]. DNICs can serve as NO storage and transport in biological systems [8–11]. In 2005, the protein-bound DNICs structure known as the Fe(NO)$_2$ unit in the active site of human glutathione transferase (GST P1-1) was reported [5,8,12,13]. DNICs chemistry is very important in the field of biology, as well as bioinorganic chemistry. In addition, some NO bound structures have also been reported in nitric oxide reductase [14–16].

Moreover, NO itself is a redox active molecule [17,18]. Therefore, it is very difficult to define its oxidation state. Due to the complicated electronic structures of transition-metal complexes with NO, the oxidation state of the metal NO unit is typically indicated using the Enemark–Feltham notation, in which the metal nitrosyl is denoted by {M(NO)$_x$}$^n$. Here, $x$ is the number of nitrosyl ligand(s) and $n$ is the sum of the metal-d and NO-$\pi$* electrons [19].

In the reported mononuclear DNICs, both the {Fe(NO)$_2$}$^9$ and {Fe(NO)$_2$}$^{10}$ oxidation states are commonly encountered [20–31]. Regarding the electronic structure of the {Fe(NO)$_2$} core in DNICs, the reported stable mononuclear DNICs can be classified into two groups: paramagnetic {Fe(NO)$_2$}$^9$ core ($S_t = 1/2$) and diamagnetic {Fe(NO)$_2$}$^{10}$ core ($S_t = 0$). From other experimental and calculation studies, both states have also been identified by their stretching frequencies ($\nu$(N–O) (>1700 cm$^{-1}$ in {Fe(NO)$_2$}$^9$ versus <1700 cm$^{-1}$ in {Fe(NO)$_2$}$^{10}$) as well as $\nu$(Fe–NO) (~550 cm$^{-1}$ in {Fe(NO)$_2$}$^9$ versus ~600 cm$^{-1}$ in {Fe(NO)$_2$}$^{10}$) as listed in Table 1 [20–31]. The well-characterized {Fe(NO)$_2$}$^{9/10}$ cores are [Fe(NO)$_2$(dmp)]$^{0/-}$ where dmp = 2,9-dimethyl-1,10-phenanthroline [20] and

[Fe(NO)$_2$(Ar-nacnac)]$^{0/-}$ is ligated by one of the β-diketoiminate ligands, where Ar-nacnac is the anion of [(2,6-diisopropylphenyl)NC(Me)]$_2$CH [21–23].

**Table 1.** Structural parameters of selected {Fe(NO)$_2$}$^9$ and {Fe(NO)$_2$}$^{10}$ DNICs and the related {FeNO}$^8$ complex.

| Complex | ν (N–O)/cm$^{-1}$ | M–N(O)/Å | N–O/Å | M–N–O/° | Reference |
|---|---|---|---|---|---|
| Four-coordinate {Fe(NO)$_2$}$^9$ | | | | | |
| [Fe(NO)$_2$(L1″)](BF$_4$) | 1831 (CH$_2$Cl$_2$) | 1.692(4) | 1.168(6) | 160.4(5) | this work |
| | 1759 (CH$_2$Cl$_2$) | 1.699(5) | 1.156(6) | 165.1(4) | |
| [Fe(NO)$_2$(dmp)](OTf) [a] | 1840 (KBr) | 1.674(6) | 1.177(7) | 170.7(6) | [20] |
| | 1746 (KBr) | 1.675(6) | 1.174(7) | 168.3(6) | |
| [Fe(NO)$_2$(Ar-nacnac)] [a] | 1761 (C$_6$D$_6$) | 1.6984(18) | 1.177(2) | 162.7(2) | [21–23] |
| | 1709 (C$_6$D$_6$) | 1.6882(18) | 1.174(2) | 170.1(2) | |
| [Fe(NO)$_2$(PPh$_3$)$_2$](PF$_6$) | 1814 (CH$_2$Cl$_2$) | 1.661(4) [b] | 1.160(6) [b] | 166.2(4) [b] | [24] |
| | 1766 (CH$_2$Cl$_2$) | | | | |
| (Et$_4$N)[Fe(NO)$_2$(S-*p*-tolyl)$_2$] [a] | 1732 (KBr) | 1.7210(15) | 1.1992(19) | 166.04(14) | [25] |
| | 1691 (KBr) | 1.7096(15) | 1.1958(19) | 170.05(14) | |
| [Fe(NO)$_2$(6-Me$_3$-TPA)](ClO$_4$) [a] | 1801 (KBr) | 1.699(3) | 1.168(4) | 159.7(4) | [26] |
| | 1726 (KBr) | 1.690(3) | 1.165(4) | 162.4(4) | |
| [Fe(NO)$_2$(NHC-iPr)](BF$_4$) [a] | 1789 (THF) | na [d] | na [d] | na [d] | [27] |
| | 1733 (THF) | na [d] | na [d] | na [d] | |
| [Fe(NO)$_2$(tmeda)](PF$_6$) [a] | 1835 (KBr) | na [d] | na [d] | na [d] | [28] |
| | 1769 (KBr) | na [d] | na [d] | na [d] | |
| Four-coordinate {Fe(NO)$_2$}$^{10}$ | | | | | |
| [Fe(NO)$_2$(dmp)] [a] | 1692 (KBr) | na [d] | na [d] | na [d] | [20] |
| | 1628 (KBr) | na [d] | na [d] | na [d] | |
| (PPN)[Fe(NO)$_2$(Ar-nacnac)] [a] | 1637 (C$_6$D$_6$) | 1.668(5) | 1.191(6) | 163.2(5) | [21] |
| | 1580 (C$_6$D$_6$) | 1.649(4) | 1.218(6) | 165.1(5) | |
| [Fe(NO)$_2$(PPh$_3$)$_2$] | 1714 (CH$_2$Cl$_2$) | 1.650(7) [b] | 1.190(10) [b] | 178.2(7) [b] | [24] |
| | 1668 (CH$_2$Cl$_2$) | | | | |
| [Fe(NO)$_2$(NHC-iPr)] [a] | 1664 (THF) | 1.642(3) [b] | 1.204(3) [b] | 173.8(2) [b] | [27] |
| | 1679 (THF) | | | | |
| [Fe(NO)$_2$(sparteine)] | 1687 (THF) | 1.6501(19) | 1.206(6) | 160.1(3) | [29] |
| | 1633 (THF) | 1.6430(19) | 1.214(5) | 176.0(3) | |
| [Fe(NO)$_2$(tmeda)] [a] | 1698 (THF) | 1.639(3) | 1.188(4) | 169.9(3) | [29] |
| | 1644 (THF) | 1.637(3) | 1.197(4) | 166.9(3) | |
| [Fe(NO)(bipy)] [a] | 1684 (solid ATR) [c] | 1.652(4) | 1.183(5) | 169.0(4) | [30] |
| | 1619 (solid ATR) [c] | 1.647(4) | 1.188(5) | 166.7(4) | |
| Four-coordinate {Fe(NO)$_2$}$^8$ | | | | | |
| [Fe(NO)(L3)] | 1696 (KBr) | 1.6753(13) | 1.1865(17) | 176.76(18) | [32] |

[a] dmp = 2,9-dimethyl-1,10-phenanthroline, Ar-nacnac = anion of [(2,6-diisopropylphenyl)NC(Me)]$_2$CH, S-*p*-tolyl = 4-methylbenzenethiolate, 6-Me$_3$-TPA = tris(6-methyl-2-pyridyl-methyl)amine, NHC-iPr = 1,3-diisopropylimidazol-2-ylidene, tmeda = *N*,*N*,*N′*,*N′*-tetramethylethylenediamine, bipy = 2,2′-Bipyridine. [b] crystallographic $C_2$ symmetry. [c] ATR = attenuated total reflection. [d] not available.

In our previous works, a series of first row transition metal complexes with NO, [M(NO)(L3)] (M = Fe, Co, Ni, and Cu), were prepared, structurally characterized and ligated by the same hydrotris(pyrazolyl)borate coligand, hydrotris(3-tertiary butyl-5-isopropyl-1-pyrazolyl)borate (denoted as L3$^-$, Scheme 1 left) [32–35]. Our synthetic strategy is to use exactly the same hindered supporting ligand to make four coordinate mono-nitrosyl transition metal complexes to directly compare their electronic properties and reactivities. However, by using this hindered tridentate ligand, we do not synthesize any dinitrosyl complexes as a stable form. The reaction of the iron complex [Fe(NO)(L3)] with NO gas, we found the formation of DNIC as [Fe(NO)$_2$(L3)] with two N–O stretching vibrations at 1805 and 1732 cm$^{-1}$ in its IR spectrum. However, this obtained DNIC is not stable, [Fe(NO)(L3)] is

easily recovered upon drying the obtained brown colored solution in vacuo [32]. Therefore, the detailed structure and properties of the DNIC [Fe(NO)$_2$(L3)] cannot be characterized in detail. Thus, we explored a less hindered bidentate nitrogen-containing ligand to stabilize DNICs.

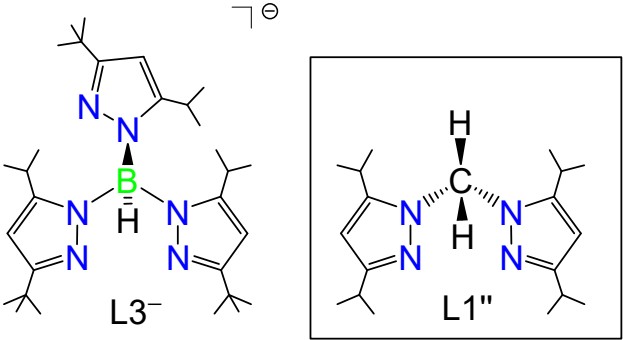

**Scheme 1.** Ligands in this research.

In this paper, we report synthesis and characterization of the dinitrosyl iron complex [Fe(NO)$_2$(L1″)](BF$_4$) by using a less hindered bis(3,5-diisopropyl-1-pyrazolyl)methane (denoted as L1″, Scheme 1 right) [36–38]. The change in the coordination number of the supporting coligands from tridentate hydrotris(pyrazolyl)borate to bidentate bis(pyrazolyl)methane can stabilize the dinitrosyl complexes. We also report the synthesis and characterization of [Co(NO)$_2$(L1″)](BF$_4$) as cobalt analogue in comparison with [Fe(NO)$_2$(L1″)](BF$_4$). In the dinitrosyl cobalt complexes, the {Co(NO)$_2$}$^{10}$ oxidation state is commonly encountered, as listed in Table 2 [23,33,39–43]. Regarding the electronic structure of {Co(NO)$_2$} core in the complexes, the reported {Co(NO)$_2$}$^{10}$ core is of diamagnetic property ($S_t = 0$) [23,33,40,41]. Moreover, there is the relatively wide range $\nu$(N–O) frequency from 1900 cm$^{-1}$ to 1700 cm$^{-1}$ [23,39,40]. For the analysis of the above two dinitrosyl complexes, we also prepared [FeCl$_2$(L1″)], [Co($\kappa^2$-O$_2$NO)$_2$(L1″)], and [Co($\kappa^2$-O$_2$N)$_2$(L1″)]. Moreover, we explored the dioxygen reactivities of [Fe(NO)$_2$(L1″)](BF$_4$) and [Co(NO)$_2$(L1″)](BF$_4$).

**Table 2.** Structural parameters of selected {Co(NO)$_2$}$^{10}$ dinitrosyl cobalt complexes and the related {CoNO}$^9$ complex.

| Complex | $\nu$ (N–O)/cm$^{-1}$ | M–N(O)/Å | N–O/Å | M–N–O/° | Reference |
|---|---|---|---|---|---|
| Four-coordinate {Co(NO)$_2$}$^{10}$ | | | | | |
| [Co(NO)$_2$(L1″)](BF$_4$) | 1875 (CH$_2$Cl$_2$) | 1.654(5) | 1.151(7) | 175.8(7) | This work |
| | | 1.673(6) | 1.137(10) | 164.6(7) | |
| | 1798 (CH$_2$Cl$_2$) | 1.673(6) | 1.137(10) | 164.6(7) | |
| | | 1.673(6) | 1.137(10) | 164.6(7) | |
| [Co(NO)$_2$(py)$_2$](BF$_4$) | 1876 (CH$_2$Cl$_2$) | 1.654(6) | 1.130(8) | 170.2(6) | [39] |
| | 1798 (CH$_2$Cl$_2$) | 1.644(6) | 1.156(8) | 170.1(6) | |
| (Et$_4$N)[Co(NO)$_2$(SPh)$_2$] [a] | 1769 (THF) | 1.684(3) | 1.106(3) | 161.4(3) | [40] |
| | 1699 (THF) | 1.651(3) | 1.126(3) | 174.9(2) | |
| [Co(NO)$_2$(Ar-nacnac)] [a] | 1801 (C$_6$D$_6$) | 1.633(3) | 1.165(5) | 173.0(4) | [23] |
| | 1706 (C$_6$D$_6$) | 1.695(4) | 1.166(5) | 150.1(4) | |
| [Co(NO)$_2$(tmeda)](BPh$_4$) [a] | 1866 (KBr) | 1.6630(12) | 1.1475(16) | 168.53(13) | [33,41] |
| | 1789 (KBr) | 1.6636(12) | 1.1582(15) | 165.83(11) | |
| Four-coordinate {CoNO}$^9$ | | | | | |
| [Co(NO)(L3)] | 1732 (KBr) | 1.700(2) | 1.112(3) | 176.1(3) | [33] |
| [Co(NO)(L0)] [a,b] | 1732 (KBr) | 1.625(5) | 1.161(6) | 173.5(6) | [42] |
| | | 1.628(5) | 1.167(6) | 175.5(6) | |
| [Co(NO)(L2)] [a] | 1732 (KBr) | 1.671(7) | 1.071(9) | 180.000(3) | [43] |

[a] SPh = benzenethiolate, Ar-nacnac = anion of [(2,6-diisopropylphenyl)NC(Me)]$_2$CH, tmeda = *N*,*N*,*N*′,*N*′-tetramethylethylenediamine, L0 = anion of hydrotris(3,5-dimethyl-1-pyrazolyl)borate, L2 = anion of hydrotris(3-tersiary butyl-5-methyl-1-pyrazolyl)borate. [b] Two independent molecule existed.

## 2. Results and Discussion

### 2.1. Synthesis

In our previous synthetic method of [Fe(NO)(L3)], we used [Fe(NO)$_2$($\mu$-I)]$_2$ as an iron and NO sources. In this work, we followed the same method to synthesize the dinitrosyl iron complex. [Fe(NO)$_2$(L1″)](BF$_4$) was prepared by the reaction of [Fe(NO)$_2$($\mu$-I)]$_2$ with L1″ and AgBF$_4$ (Scheme 2, top). In this case, one iodine ion should be removed by AgBF$_4$. If not, we do not achieve any purified complexes. The same synthetic strategy was also applied to make [Co(NO)$_2$(L1″)](BF$_4$) from [Co(NO)$_2$($\mu$-I)]$_n$ (Scheme 2, bottom). Lippard and co-workers reported the synthesis of [Co(NO)$_2$(Ar-nacnac)] from a similar salt-metathesis reaction employing [Co(NO)$_2$($\mu$-Cl)]$_2$ [23].

**Scheme 2.** Synthesis of [Fe(NO)$_2$(L1″)](BF$_4$) and [Co(NO)$_2$(L1″)](BF$_4$).

For detailed characterizations and the dioxygen reaction product analyses of the above two dinitrosyl complexes, [FeCl$_2$(L1″)], [Co($\kappa^2$-O$_2$NO)$_2$(L1″)], and [Co($\kappa^2$-O$_2$N)$_2$(L1″)] were also prepared (Scheme 3). From the metal sources of anhydrous FeCl$_2$ and Co(NO)$_3$·6H$_2$O, we obtained [FeCl$_2$(L1″)] and [Co($\kappa^2$-O$_2$NO)$_2$(L1″)], respectively. The cobalt(II) nitrito complex [Co($\kappa^2$-O$_2$N)$_2$(L1″)] can be obtained by the reaction of [CoCl$_2$(L1″)] with NaNO$_2$. All complexes are characterized by X-ray crystallography.

**Scheme 3.** Synthesis of [FeCl$_2$(L1″)], [Co($\kappa^2$-O$_2$NO)$_2$(L1″)], and [Co($\kappa^2$-O$_2$N)$_2$(L1″)].

### 2.2. Structure

The structures of [Fe(NO)$_2$(L1″)](BF$_4$) and [Co(NO)$_2$(L1″)](BF$_4$) were determined by single crystal X-ray analysis, as shown in Figure 1. The selected structural parameters are given in the caption. The small strain effect of the chelating bis(pyrazolyl)methane ligand L1″ explains the distorted tetrahedral geometry, with N11–M–N21 bond angles of 93.05(15)° (M = Fe) and 92.13(19)°

(M = Co). The most striking structural feature of the cationic $\{Fe(NO)_2\}^9$ is the differences of Fe–N(O) and N–O bond lengths. Two Fe–N(O) distances of 1.692(4) and 1.699(5) Å are within the range of 1.7210(15)–1.661(4) Å for $\{Fe(NO)_2\}^9$ and outside the range of 1.668(5)–1.637(3) Å for $\{Fe(NO)_2\}^{10}$ (Table 1) [19,20,23–29]. Two N–O distances of 1.168(6) and 1.156(6) Å are shorter than the range of 1.214(5)–1.183(5) Å for $\{Fe(NO)_2\}^{10}$ (Table 1). These structural parameters indicate that $[Fe(NO)_2(L1'')](BF_4)$ is the cationic $\{Fe(NO)_2\}^9$ dinitrosyl complex. In the case of dinitrosyl cobalt complexes, two Co–N(O) (1.654(5) and 1.673(6) Å) and two N–O (1.151(7) and 1.137(10) Å) bond lengths in $[Co(NO)_2(L1'')](BF_4)$ fall within the range of 1.684(3)–1.633(3) Å and 1.166(5)–1.106(3) Å, respectively (Table 2) [22,37–39]. To our knowledge, the cobalt dinitrosyl complexes have only $\{Co(NO)_2\}^{10}$. Only four-coordinate $\{Co(NO)\}^9$ complexes were found in $[Co(NO)(Lx)]$ ($x = 0$ and 2–3) with tridentate hydrotris(pyrazolyl)borate ligands (Table 2) [33,42,43].

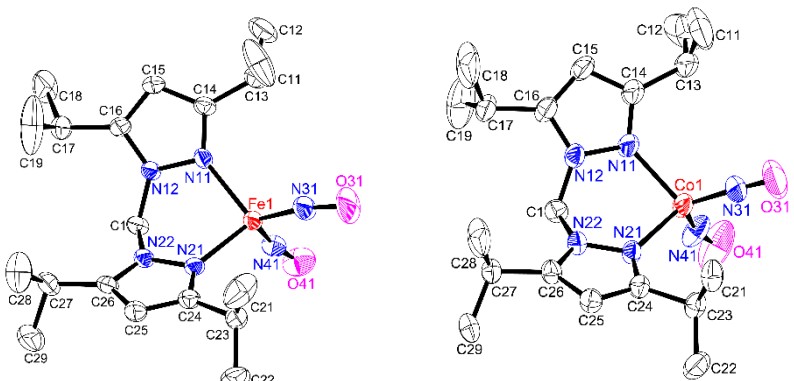

**Figure 1.** Crystal structures of $[Fe(NO)_2(L1'')]^+$ (left), $[Co(NO)_2(L1'')]^+$ (right) showing 50% displacement ellipsoids and the atom labeling scheme. $BF_4^-$ counter ion, hydrogen atoms, and solvents were omitted for reasons of clarity. Relevant bond lengths (Å) and angles (°): Fe1–N11, 2.023(4); Fe1–N21, 2.031(4); Fe1–N31, 1.692(4); Fe1–N41, 1.699(5); O31–N31, 1.168(6); O41–N41, 1.156(6); N11–Fe1–N21, 93.05(15); N11–Fe1–N31, 115.1(2); N11–Fe1–N41, 110.48(18); N21–Fe1–N31, 115.3(2); N21–Fe1–N41, 110.51(18); N31–Fe1–N41, 111.2(2); Fe1–N31–O31, 160.4(5); Fe1–N41–O41, 165.1(4). Co1–N11, 2.018(5); Co1–N21, 2.014(4); Co1–N31, 1.654(5); Co1–N41, 1.673(6); O31–N31, 1.151(7); O41–N41, 1.137(10); N11–Co1–N21, 92.13(19); N11–Co1–N31, 113.2(2); N11–Co1–N41, 110.4(2); N21–Co1–N31, 111.2(2); N21–Co1–N41, 111.9(3); N31–Co1–N41, 115.7(3); Co1–N31–O31, 175.8(7); Co1–N41–O41, 164.6(7).

In $[Fe(NO)_2(L1'')](BF_4)$ complex, the dihedral angle between N11–Fe1–N21 plane/N31–Fe1–N41 plane is 89.93°. Moreover, the distance between Fe and N11–N21–N31 plane and the distance between Fe and N11–N21–N41 plane are 0.668 Å and 0.759 Å, respectively. In $[Co(NO)_2(L1'')](BF_4)$ complex, the dihedral angle between N11–Co1–N21 plane/N31–Co1–N41 plane is 88.67°. The distance between Co and N11–N21–N31 plane and the distance between Co and N11–N21–N41 plane are 0.723 Å and 0.745 Å, respectively. Therefore, both coordination geometry of $[Fe(NO)_2(L1'')](BF_4)$ and $[Co(NO)_2(L1'')](BF_4)$ is a distorted tetrahedral.

A single crystal of $[Fe(NO)_2(L1'')](BF_4)$ contains $BF_4^-$ as a counter ion (Figure S1). Moreover, that of $[Co(NO)_2(L1'')](BF_4)$ contains $BF_4^-$ as a counter ion and tetrahydrofuran as solvents of crystallization (Figure S2). From the crystal structure of $[Fe(NO)_2(L1'')](BF_4)$, one of the NO molecules and the fluorine atom of $BF_4^-$ (N41⋯F52 = 3.348(6) and O41⋯F51 = 3.334(6) Å) and the methylene group of L1″ and the fluorine atom of $BF_4^-$ (C1⋯F51 = 3.256(6), C1⋯F52 = 3.193(6), and C1⋯F53 = 3.167(6) Å) have some interactions. In the case of $[Co(NO)_2(L1'')](BF_4)$, one of the NO molecules and the fluorine atom of $BF_4^-$ (N41⋯F51 = 3.183(11) and O41⋯F52 = 3.154(10) Å) and the methylene group of L1″ and the oxygen atom of tetrahydrofuran (C1⋯O61 = 3.106(10) Å) also have some interactions. The stability of both dinitrosyl complexes and complicated solid-state IR spectra measured by KBr pellets seem to be related to these interactions.

In [FeCl$_2$(L1″)], the iron(II) ion is four-coordinate with L1″ ligand and two chloride ions bound (Figure 2, top). The average Fe–Cl bond distance of 2.2416(6) Å is slightly longer than the Fe–Cl bond distance of 2.2369(6) Å in the four-coordinate chloride complex [FeCl(L3)] with the tridentate hydrotris(pyrazolyl)borate ligand as shown in Scheme 1 [32]. The averaged Fe–Npz bond length of 2.1073(13) Å in [FeCl$_2$(L1″)] is slightly longer than that of 2.027(4) Å in [Fe(NO)$_2$(L1″)](BF$_4$). A single crystal of [FeCl$_2$(L1″)] contains acetone as solvents of crystallization (Figure S3). Some interactions between the methylene group of L1″ and the oxygen atom of acetone (C1···O41 = 3.029(2) Å) were found.

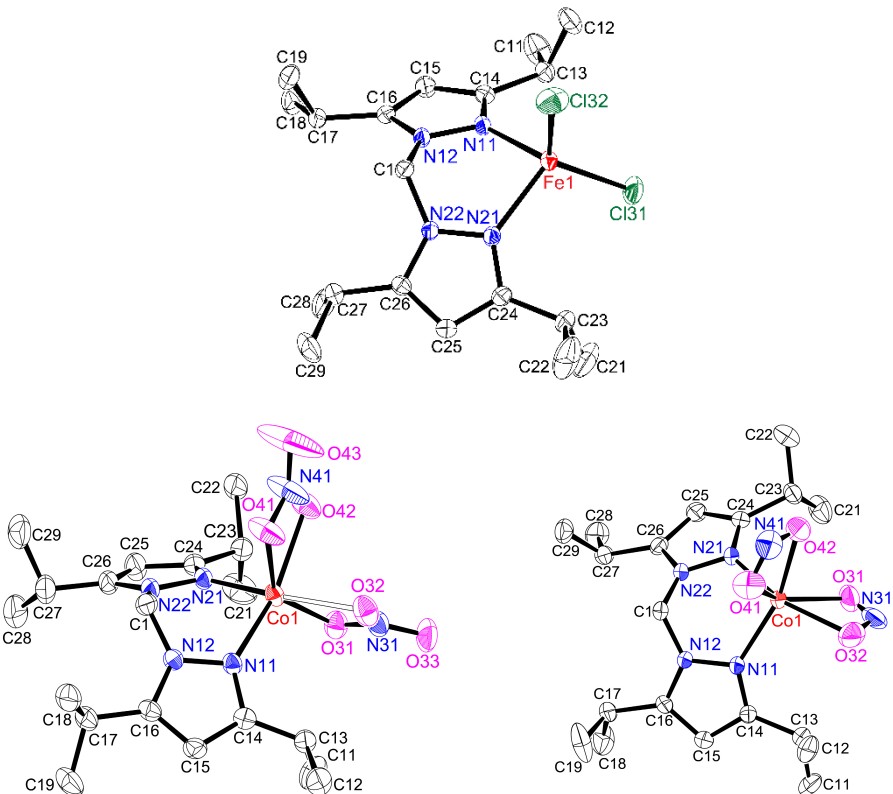

**Figure 2.** Crystal structures of [FeCl$_2$(L1″)] (top), [Co($\kappa^2$-O$_2$NO)$_2$(L1″)] (bottom, left), and [Co($\kappa^2$-O$_2$N)$_2$(L1″)] (bottom, right) showing 50% displacement ellipsoids and the atom labeling scheme. Hydrogen atoms and solvents were omitted for reasons of clarity. Relevant bond lengths (Å) and angles (°): Fe1–N11, 2.1120(13); Fe1–N21, 2.1025(9); Fe1–Cl31, 2.2445(6); Fe1–Cl32, 2.2387(6); N11–Fe1–N21, 89.45(4); N11–Fe1–Cl31, 111.05(4); N11–Fe1–Cl32, 108.66(4); N21–Fe1–Cl31, 110.35(4); N21–Fe1–Cl32, 109.32(3); Cl31–Fe1–Cl32, 122.91(2). Co1–N11, 2.0827(19); Co1–N21, 2.058(2); Co1–O31, 2.0513(19); Co1–O32, 2.265(2); Co1–O41, 2.061(2); Co1–O42, 2.248(2); O31–N31, 1.286(3); O32–N31, 1.252(3); O33–N31, 1.222(3); O41–N41, 1.299(4); O42–N41, 1.249(4); O43–N41, 1.211(4); O31–Co1–O41, 137.90(9); O31–Co1–O42, 87.96(8); O31–Co1–N11, 110.70(8); O31–Co1–N21, 105.73(8); O41–Co1–O42, 59.87(9); O41–Co1–N11, 101.35(9); O41–Co1–N21, 100.57(9); O42–Co1–N11, 160.71(8); O42–Co1–N21, 89.21(8); N11–Co1–N21, 90.26(8); Co1–O31–N31, 96.97(15); Co1–O41–N41, 95.78(18); Co1–O42–N41, 88.63(18); Co1–N11–N12, 117.24(14). Co1–N11, 2.091(2); Co1–N21, 2.071(2); Co1–O31, 2.081(3); Co1–O32, 2.235(3); Co1–O41, 2.075(3); Co1–O42, 2.251(3); O31–N31, 1.108(4); O32–N31, 1.272(5); O41–N41, 1.198(5); O42–N41, 1.254(4); O31–Co1–O32, 55.78(11); O31–Co1–O41, 139.27(11); O31–Co1–O42, 92.75(10); O31–Co1–N11, 110.44(10); O31–Co1–N21, 102.67(10); O32–Co1–O41, 93.62(12); O32–Co1–O42, 87.77(9); O32–Co1–N11, 99.92(9); O32–Co1–N21, 158.16(11); O41–Co1–O42, 56.49(11); O41–Co1–N11, 100.07(10); O41–Co1–N21, 103.23(10); O42–Co1–N11, 155.99(10); O42–Co1–N21, 90.04(9); N11–Co1–N21, 90.81(9); Co1–O31–N31, 100.2(3); Co1–O32–N31, 87.6(2); Co1–O41–N41, 100.1(2); Co1–O42–N41, 89.7(2).

We also determined the crystal structures of [Co($\kappa^2$-O$_2$NO)$_2$(L1″)] (Figure 2, bottom left) and [Co($\kappa^2$-O$_2$N)$_2$(L1″)] (Figure 2, bottom right). In both cases, the asymmetric binding modes of nitrate and nitrite are observed. [Co($\kappa^2$-O$_2$NO)$_2$(L1″)] contains the bidentate nitrate ligand with Co–O

distances: Co1–O31 = 2.0513(19) Å, Co1–O32 = 2.265(2) Å (ΔCo–O = 0.214 Å) and Co1–O41 = 2.061(2) Å, Co1–O42 = 2.248(2) Å (ΔCo–O = 0.187 Å). In [Co($\kappa^2$-O$_2$N)$_2$(L1″)], the corresponding Co–O bond lengths are Co1–O31 = 2.081(3) Å, Co1–O32 = 2.235(3) Å (ΔCo–O = 0.154 Å) and Co1–O41 = 2.075(3) Å, Co1–O42 = 2.251(3) Å (ΔCo–O = 0.176 Å). The former differences in Co1–O31 and Co1–O32 distances (ΔCo–O = 0.214 Å) are greater, and therefore, a hollow stick is used for the Co1–O32 bond in Figure 2, bottom left. Both structures are a hexa-coordinate. Since both complexes are neutral, both oxidation states of the cobalt ions are +II. Moreover, a single crystal of [Co($\kappa^2$-O$_2$N)$_2$(L1″)] contains tetrahydrofuran as solvents of crystallization (Figure S4) showing some interactions between the methylene group of L1″ and the oxygen atom of tetrahydrofuran (C1···O51 = 3.400(4) Å).

## 2.3. IR Spectroscopy

IR spectra of both dinitrosyl complexes were measured using KBr pellets as shown in Figure S5. Two isotopically sensitive bands were observed at 1822 and 1757 cm$^{-1}$ for [Fe(NO)$_2$(L1″)](BF$_4$) and at 1867 and 1800 cm$^{-1}$ for [Co(NO)$_2$(L1″)](BF$_4$). These peaks are not so clear, and some additional peaks are also observed. This comes from some interactions between NO and BF$_4^-$ (vide supra). Therefore, IR spectra were measured by solution CH$_2$Cl$_2$ as shown in Figure 3.

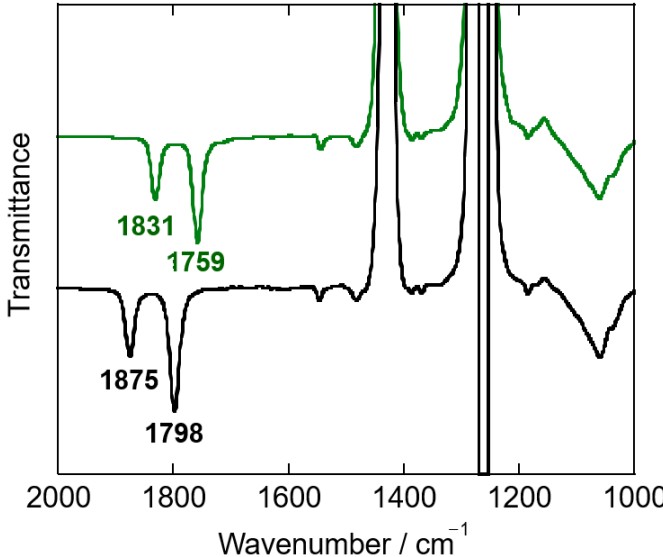

**Figure 3.** IR spectra of [Fe(NO)$_2$(L1″)](BF$_4$) (green, top) and [Co(NO)$_2$(L1″)](BF$_4$) (black, bottom) in solution (CH$_2$Cl$_2$).

IR spectrum of [Fe(NO)$_2$(L1″)](BF$_4$) exhibits symmetric $\nu_s$(N–O) and asymmetric $\nu_{as}$(N–O) frequencies at 1831 cm$^{-1}$ and 1759 cm$^{-1}$, respectively, which are similar to those for other cationic {Fe(NO)$_2$}$^9$ DNICs (see Table 1). As mentioned in the introduction, these values are consistent with the reported {Fe(NO)$_2$}$^9$ DNICs as $\nu$(N–O) > 1700 cm$^{-1}$. These frequencies were clearly shifted to 1753 cm$^{-1}$ (Δ = 78 cm$^{-1}$) and 1685 cm$^{-1}$ (Δ = 74 cm$^{-1}$) upon $^{15}$N$^{18}$O substitution (Figure S6). The magnitudes of these $^{15}$N$^{18}$O shifts are in excellent agreement with the expected shift for a diatomic harmonic oscillator about ~80 cm$^{-1}$ for $\nu$($^{14/15}$N–$^{16/18}$O). Moreover, [Co(NO)$_2$(L1″)](BF$_4$) has symmetric $\nu_s$(N–O) and asymmetric $\nu_{as}$(N–O) bands at 1875 cm$^{-1}$ and 1798 cm$^{-1}$, respectively. These bands clearly shift to 1796 cm$^{-1}$ and 1722 cm$^{-1}$ (Δ$\nu$(N–O) = 79 cm$^{-1}$ and 76 cm$^{-1}$) in [Co($^{15}$N$^{18}$O)$_2$(L1″)](BF$_4$) upon $^{15}$N$^{18}$O substitution (Figure S6). All observed isotopic shifts are consistent with the expected values calculated for a diatomic harmonic oscillator. These frequencies are very similar with those in the reported {Co(NO)$_2$}$^{10}$ (Table 2). The $\nu$(N–O) values in [Co(NO)$_2$(L1″)](BF$_4$) are higher than those in [Fe(NO)$_2$(L1″)](BF$_4$), which is consistent with the related (N–O) bond lengths.

The $\nu$(N–O) frequencies of [Fe(NO)$_2$(L1″)](BF$_4$) are compared with those of the previously reported [Fe(NO)$_2$(L3)], which was obtained by the reaction of [Fe(NO)(L3)] with NO gas in solution

as mentioned in the introduction [32]. The reaction product [Fe(NO)$_2$(L3)] shows two $\nu$(N–O) vibrations at 1805 and 1732 cm$^{-1}$. Both values are lower than those of [Fe(NO)$_2$(L1″)](BF$_4$) (1831 and 1759 cm$^{-1}$) by ~27 cm$^{-1}$. This difference comes from the coordination number of the ligand, tridentate hydrotris(pyrazolyl)borate versus bidentate bis(pyrazolyl)methane. From this observation, [Fe(NO)$_2$(L3)] would keep the five-coordinate structure even in the solution. In other words, the L3$^-$ ligand would act as $k^3$-N ligand in [Fe(NO)$_2$(L3)].

In the Fe–N(O) region, the reported [Fe(NO)$_2$(dmp)](OTf) exhibits two isotopically sensitive asymmetric $\nu_{as}$(Fe–N(O)) and symmetric $\nu_s$(N–O) bands at 581 and 534 cm$^{-1}$, which are shifted to 565 and 534 cm$^{-1}$, respectively, by nuclear resonance vibrational spectroscopy (NRVS) [20]. Moreover, [Fe(NO)(L3)] has one Fe–N(O) stretching vibration at 554 cm$^{-1}$ by far-IR spectroscopy and 550 cm$^{-1}$ by NRVS, which is shifted by 9 cm$^{-1}$ to lower energy with $^{15}$NO [32]. The far-IR spectrum of [Fe(NO)$_2$(L1″)](BF$_4$) is very complicated, however, has two isotopically sensitive asymmetric $\nu_{as}$(Fe–N(O)) and symmetric $\nu_s$(Fe–N(O)) bands at 553 and 538 cm$^{-1}$, which are shifted to 548 and 522 cm$^{-1}$, respectively (Figure S7). [Co(NO)$_2$(L1″)](BF$_4$) has also two isotopically sensitive asymmetric and symmetric bands at 640 and 592 cm$^{-1}$, which are shifted to 613 and 585 cm$^{-1}$, respectively (Figure S8). This difference is consistent with the M–N(O) bond lengths; Co–N(O) bond lengths (Co1–N31, 1.654(5) and Co1–N41, 1.673(6) Å) in [Co(NO)$_2$(L1″)](BF$_4$) are shorter than Fe–N(O) bond lengths (Fe1–N31, 1.692(4) and Fe1–N41, 1.699(5) Å) in [Fe(NO)$_2$(L1″)](BF$_4$).

The far-IR spectrum of [FeCl$_2$(L1″)] shows two intense bands at 359 and 311 cm$^{-1}$, which are assigned to $\nu$(Fe–Cl) (Figure S9). The related copper(II) complex [CuCl$_2$(L1″)] only has one broad band at 315 cm$^{-1}$ [37]. Both the cobalt(II) nitrato and nitrito complexes, [Co($\kappa^2$-O$_2$NO)$_2$(L1″)] and [Co($\kappa^2$-O$_2$N)$_2$(L1″)], are in asymmetric binding mode, as described above. The assignment of $\nu$(N–O) bands are based on the related copper(II) complexes [Cu($\kappa^2$-O$_2$NO)$_2$(L1″)] (1346 and 1182 cm$^{-1}$) [36] and [Cu($\kappa^2$-O$_2$N)$_2$(L1″)] (1505 and 1267 cm$^{-1}$) [38]. From the IR spectra (see Figure 5) $\nu$(N–O) bands are observed at 1503 and 1281 cm$^{-1}$ in [Co($\kappa^2$-O$_2$NO)$_2$(L1″)] and 1397 and 1278 cm$^{-1}$ in [Co($\kappa^2$-O$_2$N)$_2$(L1″)].

## 2.4. UV–Vis Spectroscopy

UV–Vis spectra (dichloromethane solution) and diffuse reflectance (DR) spectra (solid state) of [Fe(NO)$_2$(L1″)](BF$_4$) and [Co(NO)$_2$(L1″)](BF$_4$) are shown in Figures S10 and S11, respectively. The similarity of UV–Vis and DR spectra of [Fe(NO)$_2$(L1″)](BF$_4$) and [Co(NO)$_2$(L1″)](BF$_4$) indicates that these complexes retain their structures in the solution state compared to their crystal structures. UV–Vis spectrum of [Fe(NO)$_2$(L1″)](BF$_4$) exhibits small absorption bands at 619 (100) and 982 nm (130 M$^{-1}\cdot$cm$^{-1}$). These visible region absorption bands could be assigned to CT bands between d-electrons of iron center and dinitrosyl ligands, since [FeCl$_2$(L1″)] has no absorption bands around this region (Figure S12). This assignment is confirmed by the previous results of {Fe(NO)$_2$}$^9$ DNICs: [Fe(NO)$_2$(dmp)](OTf) at 666 (~100) and 1062 nm (~100 M$^{-1}\cdot$cm$^{-1}$) [20], (Et$_4$N)[Fe(NO)$_2$(S-*p*-tolyl)$_2$] at 795 nm (560 M$^{-1}\cdot$cm$^{-1}$) [25], and [Fe(NO)$_2$(6-Me$_3$-TPA)](ClO$_4$) at 460 (sh, 360) and 820 (200 M$^{-1}\cdot$cm$^{-1}$) [26].

[Co(NO)$_2$(L1″)](BF$_4$) has also small absorption bands at 432 (~200) and 640 nm (~100 M$^{-1}\cdot$cm$^{-1}$). These bands also could be assigned to CT bands between cobalt center and dinitrosyl ligand, since [Co($\kappa^2$-O$_2$NO)$_2$(L1″)] and [Co($\kappa^2$-O$_2$N)$_2$(L1″)] have no absorption bands around this region (Figure S13). This assignment is also confirmed by the previous results of other {Co(NO)$_2$}$^{10}$ complexes: [Co(NO)$_2$(Ar-nacnac)] at 415 (sh) and 570 nm (sh) and [Co(NO)$_2$(tmeda)](BPh$_4$) at 640 nm (~180 M$^{-1}\cdot$cm$^{-1}$).

## 2.5. Magnetic Property

The dinitrosyl cobalt complex [Co(NO)$_2$(L1″)](BF$_4$) has a sharp, well-resolved $^1$H-NMR spectrum at room temperature in acetone-$d_6$ as shown in Figure 4. Therefore, [Co(NO)$_2$(L1″)](BF$_4$) is diamagnetic. This diamagnetic magnetic property of other {Co(NO)$_2$}$^{10}$ dinitrosyl complexes is observed in (Et$_4$N)[Co(NO)$_2$(SPh)$_2$] [40], [Co(NO)$_2$(Ar-nacnac)] [23], and [Co(NO)$_2$(tmeda)](BPh$_4$) [33,41]. The origin of this diamagnetism has not been noted, to our best knowledge. However, from the

detailed computational calculations of the mononitrosyl cobalt complexes $\{Co(NO)\}^9$, we can see that this magnetism comes from the strongly antiferromagnetic interaction between d-electrons of the cobalt(II) center ($S_{Co}$ = 3/2) and the $\pi^*$ electrons of the triplet $^3NO^-$ ($S_{NO}$ = 1), resulting in a total spin for the complex of $S_t$ = 1/2 in [Co(NO)(L3)] [33] and [Co(NO)(L0)] [42]. This suggests that the diamagnetic property in $\{Co(NO)_2\}^{10}$ dinitrosyl complex [Co(NO)$_2$(L1″)](BF$_4$) can be described as a high-spin cobalt(II) center ($S_{Co}$ = 3/2) antiferromagnetically coupled to an overall (NO)$_2^-$ ($S_{NO}$ = 3/2) or a high-spin cobalt(III) center ($S_{Co}$ = 2) antiferromagnetically coupled to two $^3NO^-$ (each with $S_{NO}$ = 1). On the other hand, the preliminary EPR spectrum of [Fe(NO)$_2$(L1″)](BF$_4$) shows an isotropic signal g = 2.07 at 130 K, which is the characteristic g value of $\{Fe(NO)_2\}^9$ DNICs, showing $S_t$ = 1/2 ground state (Figure S14).

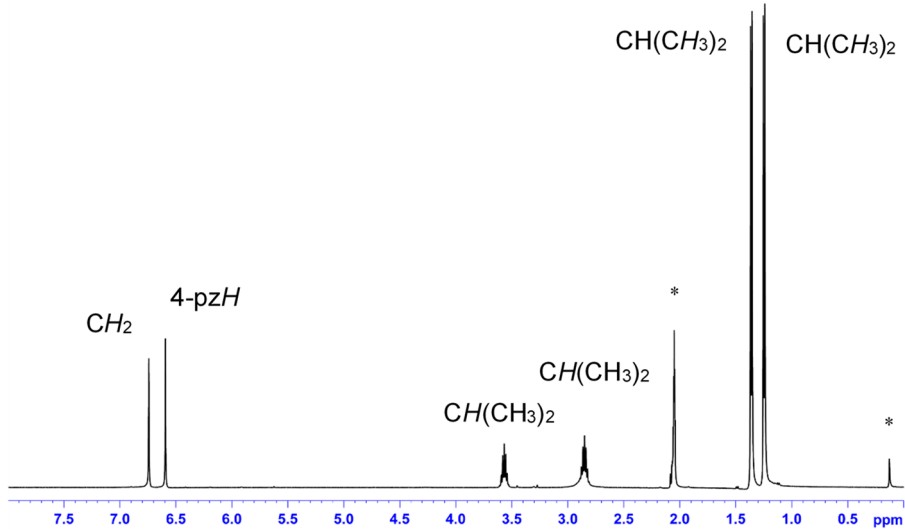

**Figure 4.** $^1$H-NMR spectrum of [Co(NO)$_2$(L1″)](BF$_4$) at room temperature in (CD$_3$)$_2$CO (* marks solvent and TMS peaks).

The paramagnetic $^1$H-NMR spectrum of [FeCl$_2$(L1″)] shows resonances in the range from 52.5 to −7.24 ppm (Figure S15). In general, the protons closest to the paramagnetic metal ion suffer the largest chemical shift and line broadening [44,45]. In this [FeCl$_2$(L1″)], the fourth position proton of the pyrazolyl ring is observed to have the largest downfield shift to 52.5 ppm (integration of 2 protons). The $^1$H-NMR spectra of [Co($\kappa^2$-O$_2$NO)$_2$(L1″)] and [Co($\kappa^2$-O$_2$N)$_2$(L1″)] show paramagnetic shifts in the range from 49.6 to −1.03 ppm and from 46.3 to −7.05 ppm, respectively (Figures S16 and S17).

*2.6. Reactivity Toward Dioxygen*

Both dinitrosyl complexes [Fe(NO)$_2$(L1″)](BF$_4$) and [Co(NO)$_2$(L1″)](BF$_4$) are stable under an argon atmosphere at room temperature. We did not see any decompositions of [Co(NO)$_2$(L1″)](BF$_4$), however, after 3 weeks, two ν(N–O) bands of [Fe(NO)$_2$(L1″)](BF$_4$) disappeared (Figure S18).

Lippard and co-workers reported the reaction of $\{Fe(NO)_2\}^{10}$ DNIC, (Bu$_4$N)[Fe(NO)$_2$(Ar-nacnac)], with air to yield $\{Fe(NO)_2\}^9$ DNIC, [Fe(NO)$_2$(Ar-nacnac)] as one-electron oxidation reaction [21]. Contrary to this result, Kim and co-workers reported dioxygen reactivity of $\{Fe(NO)_2\}^{10}$ DNIC, [Fe(NO)$_2$(tmeda)] to give new quasi-stable species as peroxynitrite bound iron mononitrosyl species, [Fe(NO)(ONOO)(tmeda)] with new stretching frequencies at 1589 and 1805 cm$^{-1}$ [46]. We reported dioxygen reactivity of [Fe(NO)(L3)] in both the solution and solid states [32]. After the reaction of [Fe(NO)(L3)] with dioxygen, the solution color changed from green to yellow within 10 min. The formation of [Fe($\kappa^2$-O$_2$N)(L3)] was confirmed by IR, $^1$H-NMR, and X-ray crystallography as the main product, with a small amount of [Fe($\kappa^2$-O$_2$NO)(L3)]. If the reaction with dioxygen is conducted

in the solid state, [Fe($\kappa^2$-O$_2$NO)(L3)] is now the main product and a small amount of [Fe($\kappa^2$-O$_2$N)(L3)] is also formed. These reactivity patterns are in agreement with our [Co(NO)(L3)] results [33].

Now the dioxygen reactivity of [Fe(NO)$_2$(L1″)](BF$_4$) and [Co(NO)$_2$(L1″)](BF$_4$) was investigated. After the reaction of [Fe(NO)$_2$(L1″)](BF$_4$) with dioxygen for 15 min, the color of the solution changed from green to yellow. In the solid-state reaction, the reaction time needs one day to complete, during which the reaction color of the solid changed from green to yellow.

The dioxygen reaction products of [Fe(NO)$_2$(L1″)](BF$_4$) exhibits new isotopically sensitive peaks at 1551 and 1288 cm$^{-1}$, which are shifted at 1466 and 1254 cm$^{-1}$ upon $^{15}$N$^{18}$O substitution in dibromomethane solution IR (Figure S19). These new peaks appearances at 1549 and 1287 cm$^{-1}$ are also detected by the dioxygen reaction in the solid state (Figure S20). This suggests that the same product was formed in both solution and solid states. As similar to the dinitrosyl iron complex [Fe(NO)$_2$(L1″)](BF$_4$), the dioxygen reaction products from the dinitrosyl cobalt complex [Co(NO)$_2$(L1″)](BF$_4$) exhibits new isotopically sensitive peaks at 1503 and 1280 cm$^{-1}$, which are shifted at 1466 and 1247 cm$^{-1}$ upon $^{15}$N$^{18}$O substitution (Figure S21). These new peaks appearances at 1503 and 1281 cm$^{-1}$ are also detected by the dioxygen reaction in the solid state (Figure S22). From the dinitrosyl iron and cobalt complexes reactivity toward dioxygen, the same products were obtained as a single component.

In order to determine the oxidized products of dinitrosyl cobalt complex, [Co($\kappa^2$-O$_2$NO)$_2$(L1″)] and [Co($\kappa^2$-O$_2$N)$_2$(L1″)] were independently synthesized as authentic complexes. The peak position comparisons between the dioxygen reaction products of [Co(NO)$_2$(L1″)](BF$_4$) (solid and solution) and the authentic complexes ([Co($\kappa^2$-O$_2$NO)$_2$(L1″)] and [Co($\kappa^2$-O$_2$N)$_2$(L1″)]) indicated the dioxygen reaction products of [Co(NO)$_2$(L1″)](BF$_4$) in both solid and solution states might be [Co($\kappa^2$-O$_2$NO)$_2$(L1″)] because of their similarities (Figure 5). However, $^1$H-NMR spectral comparisons between the dioxygen reaction products and the authentic complexes [Co($\kappa^2$-O$_2$NO)$_2$(L1″)] and [Co($\kappa^2$-O$_2$N)$_2$(L1″)], the chemical shifts and the spectral patterns of the dioxygen reaction products are clearly different of those of the authentic complexes (Figures S16, S17, S23 and S24). From these results, the accurate identification of the oxidation products is very difficult in this research. We need more detailed research to decide what happens in this dioxygen reaction. In [Fe(NO)$_2$(L1″)](BF$_4$), the reaction rate is faster than the rate in the related dinitrosyl cobalt complex. However, we cannot also decide the dioxygen reaction products in this research.

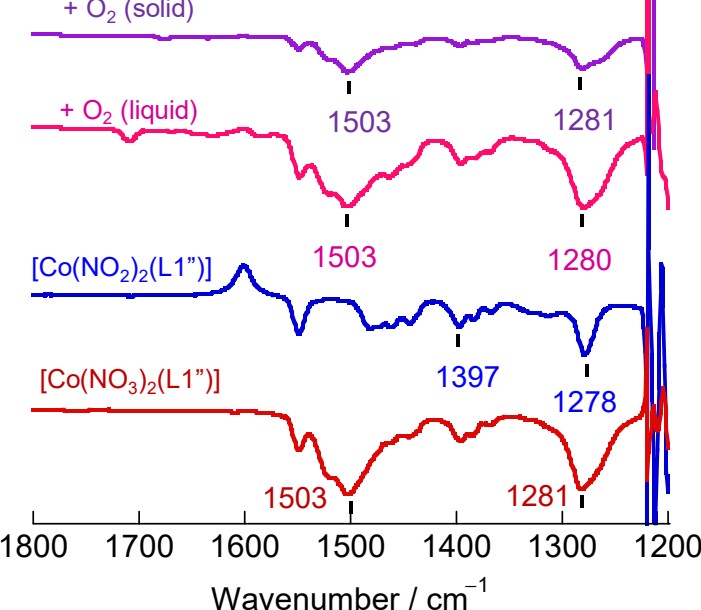

**Figure 5.** IR spectral comparisons with dioxygen reaction products of [Co(NO)$_2$(L1″)](BF$_4$) (solid and solution) and the authentic complexes ([Co($\kappa^2$-O$_2$NO)$_2$(L1″)] and [Co($\kappa^2$-O$_2$N)$_2$(L1″)]) measured by CH$_2$Br$_2$ solution.

## 3. Materials and Methods

### 3.1. Material and General Techniques

The preparation and handling of all the complexes was performed under an argon atmosphere using standard Schlenk tube techniques. Dichloromethane was purified by distillation from phosphorous pentoxide under an argon atmosphere [47]. Diethyl ether, tetrahydrofuran (THF) and *n*-heptane were distilled from sodium benzophenone ketyl under an argon atmosphere. Super-dehydrated acetone and methanol were purchased from Wako Pure Chemical Ind. Ltd. and deoxygenated by purging with argon gas. NO gas was purchased from TOKAI Holdings and purified by passing through a column filled with solid NaOH. Deuterated solvents in NMR experiments and $^{15}N^{18}O$ gas in IR/far-IR experiments were obtained from Cambridge Isotope Laboratories, Inc and Shoko Co. Ltd., respectively. $[Fe(NO)_2(\mu\text{-}I)]_2$ and $[Fe(^{15}N^{18}O)_2(\mu\text{-}I)]_2$ and were prepared as described [32,48,49]. Other reagents are commercially available and were used without further purification. L1" (bis(3,5-diisopropyl-1-pyrazolyl)methane) was prepared as reported previously [36].

### 3.2. Instrumentation

Infrared (IR) and far-IR spectra were recorded on KBr pellets (4000–400 $cm^{-1}$) using a JASCO FT/IR-6300 spectrometer (JASCO, Tokyo, Japan) and CsI pellets (650–150 $cm^{-1}$) using a JASCO FT/IR-6200 spectrometer (JASCO, Tokyo, Japan), respectively. Abbreviations used in the description of vibrational data are as follows: vs, very strong; s, strong; m, medium; w, weak. IR spectra of solution samples were obtained in thin-layer solution cells equipped with KBr windows on the same instrument. Electronic absorption (UV–Vis) spectra (dichloromethane solution and solid 240–1040 nm) were recorded on a JASCO V-570 spectrophotometer (JASCO, Tokyo, Japan). Diffuse reflectance (DR) spectra were obtained with the JASCO V-570 spectrophotometer equipped with an integrating sphere apparatus (JASCO ISN-470) using fine powder samples. $^1H$-NMR (500 MHz) spectra were obtained on a Bruker AVANCE-500 NMR spectrometer (Bruker Japan, Yokohama, Japan) at room temperature (298 K). Chemical shifts were reported as δ values relative to solvent residual signals and an internal standard (tetramethylsilane). The multiplicity of each signal is designated by the following abbreviations: s, singlet; d, doublet; sept, septet; br; broad. EPR spectra were recorded on a JEOL JES-RE2X EPR spectrometer (JEOL, Tokyo, Japan) in frozen dichloromethane solution at low temperature (−130 K) in quartz tubes (diameter 5 mm) with liquid-nitrogen temperature controller JEOL DVT UNIT. The elemental analyses (C, H, and N) were performed by the Center for Instrumental Analysis at Ibaraki University.

### 3.3. Preparation of Complexes

#### 3.3.1. $[Fe(NO)_2(L1")](BF_4)$

To a solution of $[Fe(NO)_2(\mu\text{-}I)]_2$ 0.0438 g (0.090 mmol) in dichloromethane (10 mL) was added L1" (0.0575 g, 0.182 mmol) in dichloromethane (10 mL). The resulting black green colored solution was stirred at 0 °C under an argon atmosphere for 2 h. After that, a solution containing 0.0380 g (0.195 mmol) of $AgBF_4$ in acetone (5 mL) was added dropwise and the mixture was stirred at 0 °C for 30 min. The green solution was filtered off using Celite. The solvent was removed under vacuum. Recrystallization from tetrahydrofuran/*n*-heptane at −30 °C gave dark green crystals. Single crystals suitable for X-ray diffraction were obtained by slow recrystallization under the same experimental conditions.

Yield: 81% (0.0768 g, 0.148 mmol). Anal. Calcd. for $C_{19}H_{32}BF_4FeN_6O_2$ + $H_2O$: C, 42.48; H, 6.38; N, 15.65. Found: C, 42.71; H, 6.11; N, 15.25. IR (KBr) ν/$cm^{-1}$: 3448w, 3136w, 2971s, 2935w, 2871w, 1822s ν(N–O), 1757vs ν(N–O), 1729m, 1697w, 1470w, 1396m, 1270m, 1066vs ν($BF_4$), 841w. Far-IR (CsI) ν/$cm^{-1}$: 663s, 585w, 553m, 538w, 521s, 472w, 419w, 378w, 314w, 253w, 211m. IR ($CH_2Cl_2$) ν/$cm^{-1}$: 1831s ν(N–O), 1759vs ν(N–O), 1061vs ν($BF_4$). UV–vis ($CH_2Cl_2$) λ/nm ($\varepsilon$/$M^{-1} \cdot cm^{-1}$): 341 (1500), 500 (sh, ~180),

619 (100), 982 (130). UV–vis (nujol) $\lambda$/nm: 358, 500, 598, 906. DR (solid) $\lambda$/nm: 356, 500, 606, 706, 940, 1954. EPR (solid): 2.07. EPR ($CH_2Cl_2$ solution): 2.07.

### 3.3.2. [Fe($^{15}N^{18}O$)$_2$(L1″)](BF$_4$)

The $^{15}N^{18}O$-labeled complex was prepared by the same method as the corresponding unlabeled complex using [Fe($^{15}N^{18}O$)$_2$($\mu$-I)]$_2$ (0.0734 g, 0.151 mmol) and ligand L1″ (0.0964 g, 0.305 mmol).

Yield: 86% (0.141 g, 0.262 mmol). IR (KBr) $\nu$/cm$^{-1}$: 3408w, 3136m, 2971s, 2934m, 2871w, 1745vs $\nu$($^{15}N$–$^{18}O$), 1683vs $\nu$($^{15}N$–$^{18}O$),1627w, 1545m, 1444m, 1396s, 1271s, 1066 vs $\nu$(BF$_4$), 880m. Far-IR (CsI) $\nu$/cm$^{-1}$: 663s, 585w, 562m, 547m, 522s, 471w, 426w, 408w, 388w, 369w, 306w, 261w, 225w, 208m. IR ($CH_2Cl_2$) $\nu$/cm$^{-1}$:1753s $\nu$($^{15}N$–$^{18}O$), 1685 vs $\nu$($^{15}N$–$^{18}O$), 1061 vs $\nu$(BF$_4$).

### 3.3.3. [Co(NO)$_2$($\mu$-I)]$_n$

This complex was prepared using a modified version of the reported method [49]. To remove water, the starting cobalt powder 0.995 g (16.9 mmol) was settled in an oil bath at 150 °C for 6 h. After cooling to room temperature, acetone (23 mL) was added and the mixture was stirred for a few minutes. To this solution, an ether solution (34 mL) of iodine (1.08 g, 4.26 mmol) was added dropwise over 1 h. About 200 cm$^3$ of NO gas was passed through the solution, and the mixture was stirred at room temperature for 1 h, which was then stirred for another 1 h. After the reaction, the solvent was removed under vacuum. Dark brown crystals were obtained by sublimation of the complex.

Yield: 65% (1.61 g, 6.55 mmol). IR (KBr) $\nu$/cm$^{-1}$: 1874s $\nu$(N–O), 1825vs $\nu$(N–O), 1799 vs. $\nu$(N–O). UV–vis (MeOH) $\lambda$/nm ($\varepsilon$/M$^{-1}$·cm$^{-1}$): 255 (sh), 330 (sh, 340), 502 (460), 756 (220). DR (solid, nm): 208, 354, 528, 832.

### 3.3.4. [Co($^{15}N^{18}O$)$_2$($\mu$-I)]$_n$

The $^{15}N^{18}O$-labelled complex was prepared by a procedure analogous to that used for [Co(NO)$_2$($\mu$-I)]$_n$ using iron powder (0.109 g, 1.85 mmol), iodine (0.152 g, 0.599 mmol), and $^{15}N^{18}O$ gas.

Yield: 64% (0.229 g, 0.932 mmol). IR (KBr) $\nu$/cm$^{-1}$: 1797s $\nu$($^{15}N$–$^{18}O$), 1746vs $\nu$($^{15}N$–$^{18}O$).

### 3.3.5. [Co(NO)$_2$(L1″)](BF$_4$)

To a solution of [Co(NO)$_2$($\mu$-I)]$_n$ (0.0370 g, 0.151 mmol) in dichloromethane (10 mL) L1″ (0.0478 g, 0.151 mmol) in dichloromethane (10 mL) was added. The resulting dark brown colored solution was stirred at 0 °C under an argon atmosphere for 2 h. After that, a solution containing 0.0308 g (0.158 mmol) of AgBF$_4$ in acetone (5 mL) was added and the mixture was stirred at 0 °C for 30 min. The brown solution was filtered off using Celite. The solvent was removed under vacuum. Recrystallization from tetrahydrofuran/*n*-heptane at −30 °C gave dark brown crystals. Single crystals suitable for X-ray diffraction were obtained by slow recrystallization under the same experimental conditions.

Yield: 81% (0.064 g, 0.123 mmol). Calcd for C$_{19}$H$_{32}$BF$_4$CoN$_6$O$_2$ + 0.5 H$_2$O: C, 42.96; H, 6.26; N, 15.82. Found: C; 42.95, H; 6.37, N; 15.59. IR (KBr) $\nu$/cm$^{-1}$: 3447w, 3137w, 2967s, 2933w, 2871w, 1867s $\nu$(N–O), 1800vs $\nu$(N–O), 1739m, 1620m, 1547w, 1467w, 1399w, 1385m, 1277m, 1083vs $\nu$(BF$_4$), 797w. Far-IR (CsI) $\nu$/cm$^{-1}$: 663s, 640m, 586w, 592w, 550m, 522s, 473w, 431w, 406w, 379w, 352w, 327w, 310w. IR ($CH_2Cl_2$) $\nu$/cm$^{-1}$: 1875s $\nu$(N–O), 1798vs $\nu$(N–O), 1060vs $\nu$(BF$_4$). UV–vis ($CH_2Cl_2$) $\lambda$/nm ($\varepsilon$/M$^{-1}$·cm$^{-1}$): 366 (620), 432 (sh, 200), 640 (sh). DR (solid) $\lambda$/nm: 378, 470, 652. $^1$H-NMR ((CD$_3$)$_2$CO, 500 MHz) $\delta$/ppm: 1.25 (d, 12H, *J* = 6.8 Hz, CH(C*H*$_3$)$_2$), 1.36 (d, 12H, *J* = 6.8 Hz, CH(C*H*$_3$)$_2$), 2.85 (sept, 2H, *J* = 6.8 Hz, C*H*(CH$_3$)$_2$), 3.57 (sept, 2H, *J* = 6.8 Hz, C*H*(CH$_3$)$_2$), 6.59 (s, 2H, 4-*H*(pz)), 6.74 (s, 2H, C*H*$_2$).

### 3.3.6. [Co($^{15}N^{18}O$)$_2$(L1″)](BF$_4$)

The $^{15}N^{18}O$-labeled complex was prepared by the same method as the corresponding unlabeled complex using [Co($^{15}N^{18}O$)$_2$($\mu$-I)]$_n$ (0.0627 g, 0.255 mmol) and ligand L1″ (0.0807 g, 0.255 mmol).

Yield: 96% (0.128 g, 0.245 mmol). (KBr) $v$/cm$^{-1}$: 3464w, 2966s, 2932m, 2871m, 1811m, 1799s $v(^{15}N{-}^{18}O)$, 1731vs $v(^{15}N{-}^{18}O)$,1665m, 1548m, 1463m, 1277s, 1060 vs $v(BF_4)$, 797m. Far-IR (CsI) $v$/cm$^{-1}$: 662s, 613w, 585w, 549m, 522s, 473w, 406w, 377w, 354w, 307w. IR (CH$_2$Cl$_2$) $v$/cm$^{-1}$: 1796s $v(^{15}N{-}^{18}O)$, 1722 vs $v(^{15}N{-}^{18}O)$, 1060 vs $v(BF_4)$.

### 3.3.7. [FeCl$_2$(L1″)]

L1″ (0.268 g, 0.847 mmol) dissolved in dichloromethane (15 mL) was added to a solution of anhydrous FeCl$_2$ (0.107 g, 0.844 mmol) in dichloromethane (15 mL). After stirring overnight, the solvent was filtered off using Celite. The organic solvent was then dried over under reduced pressure. Recrystallization from acetone at −30 °C gave colorless crystals.

Yield: 86% (0.366 g, 0.730 mmol). Anal. Calcd. for C$_{19}$H$_{32}$Cl$_2$FeN$_4$ + (CH$_3$COCH$_3$). C, 52.71; H, 7.64; N, 11.18. Found: C, 52.44; H, 7.48; N, 11.16. IR (KBr) $v$/cm$^{-1}$: 3615m, 3534m, 2967s, 2932m, 2871w, 1708s $v(C{=}O)$, 1547s, 1471m, 1338m, 1396m, 1385m, 1281s, 1226w, 1185m, 1041m, 1023w, 1066s, 811m, 791m. Far–IR (CsI) $v$/cm$^{-1}$: 674w, 661m, 584w, 531m, 507w, 472w, 407w, 359vs, 311s, 250m. UV–vis (CH$_2$Cl$_2$) $\lambda$/nm ($\varepsilon$/M$^{-1}\cdot$cm$^{-1}$): 307 (100), 365 (70). DR (solid) $\lambda$/nm: 308, 1536, 1770. $^1$H-NMR (CDCl$_3$, 500 MHz) $\delta$/ppm: −7.24 (br, 12H, CH(CH$_3$)$_2$), 2.18 (s, 6H, CH$_3$(acetone)), 4.81 (br, 12H, CH(CH$_3$)$_2$), 21.9 (s, 2H, H$_2$C), 52.5 (s, 2H, 4-H(pz)) (Methine protons of the isopropyl group were not observed.).

### 3.3.8. [Co($\kappa^2$-O$_2$NO)$_2$(L1″)]

To a solution of Co(NO$_3$)$_2$·6H$_2$O (0.0824 g, 0.283 mmol) in acetone (10 mL) was added L1″ (0.0962 g, 0.304 mmol) dissolved in acetone (10 mL). The color of the solution gradually turned deep red violet. After having been stirred for 2 h, the solvent was removed under vacuum. The residue was extracted with dichloromethane (20 mL), and undissolved powder was filtered off using Celite. The solution was then concentrated by the removal of solvent under reduced pressure. Recrystallization from dichloromethane/$n$-heptane at −30 °C gave red violet crystals. Single crystals suitable for X-ray diffraction were obtained by slow recrystallization under the same experimental conditions.

Yield: 93% (0.132 g, 0.264 mmol). Anal. Calcd. for C$_{19}$H$_{32}$CoN$_6$O$_6$: C, 45.69; H, 6.46; N, 16.83. Found: C; 45.81, H; 6.52, N; 16.78. IR (KBr) $v$/cm$^{-1}$: 3437w, 2971s, 2934m, 2872w, 1549m, 1506vs, 1385vs, 1279vs, 1185m, 1166w, 1059w, 1017m, 808m. Far–IR (CsI) $v$/cm$^{-1}$: 674w, 661m, 584w, 531m, 507w, 472w, 407w, 359vs, 311s, 250m. IR (CH$_2$Cl$_2$) $v$/cm$^{-1}$: 1506s, 1397w, 1386w, 1370w, 1353w, 1058m, 1019m. IR (CH$_2$Br$_2$) $v$/cm$^{-1}$: 1549m, 1503s, 1396m, 1386w, 1368w, 1281s, 1059w, 1017m. UV–vis (CH$_2$Cl$_2$) $\lambda$/nm) ($\varepsilon$/M$^{-1}\cdot$cm$^{-1}$): 300 (sh, 900), 533 (140), 1128 (sh, ~20). $^1$H-NMR ((CD$_3$)$_2$CO, 500 MHz) $\delta$/ppm: −1.03 (br, 12H, CH(CH$_3$)$_2$), 1.25 (br, 12H, CH(CH$_3$)$_2$), 3.39 (br, 4H, CH(CH$_3$)$_2$), 6.80 (br, 2H, H$_2$C), 49.6 (s, 2H, 4-H(pz)).

### 3.3.9. [Co($\kappa^2$-O$_2$N)$_2$(L1″)]

To a solution of CoCl$_2$·6H$_2$O (0.112 g, 0.470 mmol) in acetone (10 mL) was added L1″ (0.146 g, 0.461 mmol) dissolved in acetone (10 mL). The color of the solution gradually turned deep blue. After having been stirred for 3 h, the solvent was removed under vacuum. The blue crude products as [CoCl$_2$(L1″)] were obtained. After that, to a solution of blue crude products (0.204, 0.457 mmol) in dichloromethane (10 mL) was added 2 equivalents of NaNO$_2$ (0.0637 g, 0.923 mmol) in methanol (10 mL). After having been stirred for 2 h, undissolved powder was filtered off using Celite. The filtrate was evaporated under reduced pressure to give a blue purple solid. Recrystallization from tetrahydrofuran/$n$-heptane at −30 °C gave blue violet crystals after one more Celite off. Single crystals suitable for X-ray diffraction were obtained by slow recrystallization under the same experimental conditions.

Yield: 63% (0.136 g, 0.288 mmol). Anal. Calcd. for C$_{19}$H$_{32}$CoN$_6$O$_4$ + 0.25 H$_2$O: C, 48.36; H, 6.94; N, 17.81. Found: C; 48.58, H; 6.96, N; 17.53. IR (KBr) $v$/cm$^{-1}$: 3447w, 3133w, 2970vs, 2933m, 2872w, 1627w, 1549vs, 1482s, 1444m, 1399s, 1335m, 1279vs, 1184s, 1062m, 1022w, 981w, 810m, 661w. IR (CH$_2$Cl$_2$) $v$/cm$^{-1}$: 1550m, 1465w, 1397m, 1385m, 1184s, 1059m, 1024m. IR (CH$_2$Br$_2$) $v$/cm$^{-1}$: 1549m, 1482m,

1463m, 1444m, 1397m, 1384m, 1367m, 1278s, 1058m, 1024m. UV–vis ($CH_2Cl_2$) $\lambda$/nm ($\varepsilon$/M$^{-1}\cdot$cm$^{-1}$): 298 (680), 539 (130), 615 (sh, ~60), 677 (sh), 1233 (~20). $^1$H-NMR (($CD_3$)$_2$CO, 500 MHz) $\delta$/ppm: −7.05 (br, 2H, $H_2$C), 0.65 (br, 12H, CH(C$H_3$)$_2$), 2.94 (br, 12H, CH(C$H_3$)$_2$), 4.24 (br, 2H, C$H$(CH$_3$)$_2$), 5.62 (br, 2H, C$H$(CH$_3$)$_2$), 46.3 (s, 2H, 4-$H$(pz)).

### 3.4. O$_2$ Reactivity of the Dinitrosyl Complexes

#### 3.4.1. Reaction of [Fe(NO)$_2$(L1″)](BF$_4$) with O$_2$ in Solution

In a 50 mL Schlenk tube, [Fe(NO)$_2$(L1″)](BF$_4$) was dissolved in dichloromethane at room temperature in an argon atmosphere. The argon was then replaced with dioxygen gas and the solution was stirred at room temperature for 15 min. During this time, the color of the solution changed to yellow. After the reaction finished, the solvent was removed under vacuum. Then, the sample was used for IR measurement.

IR ($CH_2Br_2$) $\nu$/cm$^{-1}$: 1551s (shifted to 1508 upon $^{15}$N$^{18}$O substitution), 1458s, 1372m, 1288s, 1256s, 1112m, 1065m, 1015w.

#### 3.4.2. Reaction of [Fe(NO)$_2$(L1″)](BF$_4$) with O$_2$ in the Solid State

In a 50 mL Schlenk tube, [Fe(NO)$_2$(L1″)](BF$_4$) was added and the microcrystalline material was pulverized by stirring under an argon atmosphere. After that, the atmosphere was replaced with dioxygen gas. The color of the solid changed over time to yellow, which was further facilitated for 1 day. Then, the sample was used for IR measurement.

IR ($CH_2Br_2$) $\nu$/cm$^{-1}$: 1550s (shifted to 1512 upon $^{15}$N$^{18}$O substitution), 1459w, 1372w, 1287s, 1255s, 1109m, 1065m, 1013w.

#### 3.4.3. Reaction of [Co(NO)$_2$(L1″)](BF$_4$) with O$_2$ in Solution

In a 50 mL Schlenk tube, [Co(NO)$_2$(L1″)](BF$_4$) was dissolved in dichloromethane at room temperature in an argon atmosphere. The argon was then replaced with dioxygen gas and the solution was stirred at room temperature for 3 h. During this time, the color of the solution changed to purple. After the reaction finished, the solvent was removed under vacuum. Then, the sample was used for IR measurement.

IR ($CH_2Br_2$) $\nu$/cm$^{-1}$: 1549m, 1520s, 1503s (shifted to 1467 upon $^{15}$N$^{18}$O substitution), 1465s, 1395m, 1279s, 1060m, 1020w.

#### 3.4.4. Reaction of [Co(NO)$_2$(L1″)](BF$_4$) with O$_2$ in the Solid State

In a 50 mL Schlenk tube, [Co(NO)$_2$(L1″)](BF$_4$) was added and the microcrystalline material was pulverized by stirring under an argon atmosphere. After that, the atmosphere was replaced with dioxygen gas. The color of the solid changed over time to purple, which was further facilitated for 14 days. Then, the sample was used for IR measurement.

IR ($CH_2Br_2$) $\nu$/cm$^{-1}$: 1549m, 1503s, (shifted to 1467 upon $^{15}$N$^{18}$O substitution), 1396m, 1385w, 1369w, 1281s, 1066m, 1018w.

### 3.5. X-ray Crystal Structure Determination

Crystal data and refinement parameters for two dinitrosyl transition-metal complexes [Fe(NO)$_2$(L1″)](BF$_4$) and [Co(NO)$_2$(L1″)](BF$_4$) were listed in Table 3 and those of the related [FeCl$_2$(L1″)], [Co(NO$_3$)$_2$(L1″)], and [Co(NO$_2$)$_2$(L1″)] are given in Table 4. All crystallographic data have been deposited at the CCDC, 12 Union Road, Cambridge CB2 1EZ, UK and copies can be obtained on request, free of charge, by quoting the publication citation and the deposition numbers. CCDC numbers: 1944313 for [Fe(NO)$_2$(L1″)](BF$_4$), 1944314 for [Co(NO)$_2$(L1″)](BF$_4$), 1944315 for [FeCl$_2$(L1″)], 1944316 for [Co(NO$_3$)$_2$(L1″)], and 1944317 for [Co(NO$_2$)$_2$(L1″)]. Diffraction data were measured on a Rigaku XtaLAB P200 diffractometer (Rigaku, Tokyo, Japan) using multi-layer mirror monochromated

Mo Kα (λ = 0.71075 Å) radiation at −95 °C. A crystal of suitable size and quality was coated with Paratone-N oil and mounted on a Dual-Thickness MicroLoop LD (200 μM), purchased from MiTeGen. The unit cell parameters were determined using CrystalClear from 18 images [50]. The crystal to detector distance was ca. 45 mm. Data were collected using 0.5° intervals in φ and ω to a maximum 2θ value of 55.0°. The highly redundant data sets were reduced using CrystalClear (Rigaku Oxford Diffraction) [50]. An empirical absorption correction was applied for each complex. Structures were solved by direct methods (SIR2008) [51]. For the refinements, the position of the metal ions and their first coordination sphere were located from a direct method *E*-map; other non-hydrogen atoms were found in alternating difference Fourier syntheses, and least-squares refinement cycles. During the final refinement cycles the temperature factors were refined anisotropically. Refinement was carried out by a full matrix least-squares method on $F^2$. All calculations were performed with the CrystalStructure [52] crystallographic software package except for refinement, which was performed using *SHELXL* 2013 [53]. Hydrogen atoms were placed in calculated positions. Unit weightings were used. The absolute configurations of [[Fe(NO)$_2$(L1″)](BF$_4$) and [Co(NO)$_2$(L1″)](BF$_4$) were confirmed from the values of the Flack parameters (0.041(7) and 0.018(5)) refined using 1698 and 2482 Parsons' quotients pairs, respectively [54]. For [Co(NO)$_2$(L1″)](BF$_4$), PLATON SQUEEZE was used to account for severely disordered solvent molecules [55].

**Table 3.** Crystal data and structure refinement of dinitrosyl transition-metal complexes.

| Complexes | [Fe(NO)$_2$(L1″)](BF$_4$) | [Co(NO)$_2$(L1″)](BF$_4$)·thf |
|---|---|---|
| CCDC deposition number | 1944313 | 1944314 |
| Empirical Formula | C$_{19}$H$_{32}$BF$_4$FeN$_6$O$_2$ | C$_{23}$H$_{40}$BF$_4$CoN$_6$O$_3$ |
| Formula Weight | 519.15 | 539.54 |
| Crystal System | monoclinic | monoclinic |
| Space Group | *Cc* (#9) | *Cc* (#9) |
| *a*/Å | 13.093(3) | 11.777(3) |
| *b*/Å | 19.318(5) | 15.679(3) |
| *c*/Å | 10.174(3) | 17.388(4) |
| β/° | 90.349(7) | 108.515(4) |
| *V*/Å$^3$ | 2573.3(12) | 3044.5(12) |
| *Z* | 4 | 4 |
| $D_{calc}$/g·cm$^{-3}$ | 1.340 | 1.297 |
| μ(Mo Kα)/cm$^{-1}$ | 6.404 | 6.213 |
| Temperature/°C | −95 | −95 |
| 2θ range, ° | 6–55 | 6–55 |
| Reflections collected | 21004 | 24068 |
| Unique reflections | 4870 | 6765 |
| $R_{int}$ | 0.0369 | 0.0366 |
| Number of Variables | 298 | 343 |
| Refls./Para ratio | 16.34 | 19.72 |
| Residuals: *R1* (*I* > 2σ (*I*)) | 0.0473 | 0.0536 |
| Residuals: *R* (All reflections) | 0.0506 | 0.0663 |
| Residuals: *wR2* (All reflections) | 0.1365 | 0.1596 |
| Good. of fit ind. | 1.042 | 0.923 |
| Flack parameters | 0.041(7) | 0.018(5) |
| (Parsons' quotients) | 1698 | 2482 |
| Max/min peak,/e Å$^{-3}$ | 1.07/−0.34 | 1.04/−0.40 |

$R1 = \Sigma \ ||Fo| - |Fc||/\Sigma \ |Fo|, \ wR2 = [ \ \Sigma \ (w \ (Fo^2 - Fc^2)^2)/\Sigma \ w(Fo^2)^2]^{1/2}.$

**Table 4.** Crystal data and structure refinement of the related complexes.

| Complexes | [FeCl$_2$(L1″)]·(CH$_3$)$_2$CO | [Co(NO$_3$)$_2$(L1″)] | [Co(NO$_2$)$_2$(L1″)]·thf |
|---|---|---|---|
| CCDC number | 1944315 | 1944316 | 1944317 |
| Empirical Formula | C$_{22}$H$_{38}$Cl$_2$FeN$_4$O | C$_{19}$H$_{32}$Co$_6$N$_6$O$_6$ | C$_{23}$H$_{40}$CoN$_6$O$_5$ |
| Formula Weight | 501.32 | 499.43 | 539.54 |
| Crystal System | Triclinic | Orthorhombic | monoclinic |
| Space Group | $P\bar{1}$ (#2) | *Pbca* (#61) | $P2_1/c$ (#14) |
| *a*/Å | 8.9680(14) | 16.8808(19) | 10.0033(15) |
| *b*/Å | 9.8986(15) | 17.1117(18) | 17.788(3) |
| *c*/Å | 15.939(2) | 18.875(2) | 16.691(3) |
| $\alpha$/° | 99.7610(17) | — | — |
| $\beta$/° | 92.444(4) | — | 94.743(4) |
| $\gamma$/° | 107.572(4) | — | — |
| *V*/Å$^3$ | 1322.7(3) | 5452.2(10) | 2959.8(9) |
| Z | 2 | 8 | 4 |
| $D_{calc}$/g·cm$^{-3}$ | 1.259 | 1.217 | 1.211 |
| $\mu$(Mo K$\alpha$)/cm$^{-1}$ | 7.905 | 6.695 | 6.191 |
| Temperature/°C | −95 | −95 | −95 |
| 2$\theta$ range, ° | 6–55 | 6–55 | 6–55 |
| Reflections collected | 43303 | 42232 | 46976 |
| Unique reflections | 6059 | 6251 | 6795 |
| $R_{int}$ | 0.0160 | 0.0294 | 0.0824 |
| Number of Variables | 271 | 289 | 316 |
| Refls./Para ratio | 22.36 | 21.63 | 21.50 |
| Residuals: *R1* (*I* > 2 σ (*I*)) | 0.0274 | 0.0447 | 0.0539 |
| Residuals: *R* (All reflections) | 0.0293 | 0.0579 | 0.0805 |
| Residuals: *wR2* (All reflections) | 0.0755 | 0.1449 | 0.1427 |
| Good. of fit ind. | 1.054 | 1.112 | 1.069 |
| Max/min peak,/e Å$^{-3}$ | 0.44/−0.42 | 0.87/−0.55 | 0.65/−0.43 |

$R1 = \Sigma \, ||Fo| - |Fc||/\Sigma \, |Fo|, \, wR2 = [\, \Sigma \, (w \, (Fo^2 - Fc^2)^2)/\Sigma \, w(Fo^2)^2]^{1/2}.$

## 4. Conclusions

We synthesized the dinitrosyl iron and cobalt complexes, [Fe(NO)$_2$(L1″)](BF$_4$) and [Co(NO)$_2$(L1″)](BF$_4$). X-ray analysis shows that the coordination geometry is a distorted tetrahedral, which are surprisingly stable under argon atmosphere. IR spectrum of [Fe(NO)$_2$(L1″)](BF$_4$) exhibits symmetric $\nu_s$(N–O) and asymmetric $\nu_{as}$(N–O) frequencies at 1831 and 1759 cm$^{-1}$, respectively, which are similar to those for the other cationic {Fe(NO)$_2$}$^9$ dinitrosyl complexes. Moreover, [Co(NO)$_2$(L1″)](BF$_4$) has symmetric $\nu_s$(N–O) and asymmetric $\nu_{as}$(N–O) bands at 1875 and 1798 cm$^{-1}$, respectively. These values are consistent with those of the other reported {Co(NO)$_2$}$^{10}$ complexes. These symmetric and asymmetric $\nu$(N–O) frequencies are shifted upon isotopic substitutions. We also characterized $\nu$(M–N(O)) frequencies by far-IR spectroscopy. UV–vis and magnetic properties results are also consistent with the above assignments. Finally, we tried to the reactivity of both dinitrosyl complex with dioxygen. The related [Co($\kappa^2$-O$_2$NO)$_2$(L1″)] and [Co($\kappa^2$-O$_2$N)$_2$(L1″)] complexes are synthesized and characterized as authentic samples. Compared between the dioxygen reaction products and the related nitrato and nitrito complexes in their IR spectra, [Co($\kappa^2$-O$_2$NO)$_2$(L1″)] might be obtained from both solution and solid states dioxygen reactions. However, $^1$H-NMR spectra of the dioxygen products and [Co($\kappa^2$-O$_2$NO)$_2$(L1″)] are clearly different. Therefore, we cannot conclude these dioxygen products. It is very different from our mononitrosyl [Fe(NO)(L3)] results. We are now investigating the reactivity with external substrates by mononitrosyl and dinitrosyl complexes and the detailed characterization of these dinitorosyl complexes to reveal their oxidation states.

**Supplementary Materials:** The following items are available online at http://www.mdpi.com/2304-6740/7/10/116/s1, CIFs and checkCIF reports. Figure S1: Crystal structure of [Fe(NO)$_2$(L1″)](BF$_4$), Figure S2: Crystal structure of [Co(NO)$_2$(L1″)](BF$_4$)·thf, Figure S3. Crystal structure of [FeCl$_2$(L1″)]·(CH$_3$)$_2$CO. Figure S4. Crystal structure of [Co(NO$_2$)$_2$(L1″)] thf, Figure S5: IR spectra of [Fe(NO)$_2$(L1″)](BF$_4$) and [Co(NO)$_2$(L1″)](BF$_4$) (KBr), Figure S6: IR spectra of [Fe($^{15}$N$^{18}$O)$_2$(L1″)](BF$_4$) and [Co($^{15}$N$^{18}$O)$_2$(L1″)](BF$_4$) (CH$_2$Cl$_2$), Figure S7: Far-IR spectra of [Fe(NO)$_2$(L1″)](BF$_4$) and [Fe($^{15}$N$^{18}$O)$_2$(L1″)](BF$_4$), Figure S8: Far-IR spectra of [Co(NO)$_2$(L1″)](BF$_4$) and [Co($^{15}$N$^{18}$O)$_2$(L1″)](BF$_4$), Figure S9: Far-IR spectra of [FeCl$_2$(L1″)](BF$_4$) (CsI pellet), Figure S10: UV–vis and diffuse reflectance spectra of [Fe(NO)$_2$(L1″)](BF$_4$), Figure S11: UV–vis and diffuse reflectance spectra of [Co(NO)$_2$(L1″)](BF$_4$), Figure S12: UV–vis and diffuse reflectance spectra of [FeCl$_2$(L1″)], Figure S13: UV–is spectra of [Co($\kappa^2$-O$_2$NO)$_2$(L1″)] and [Co($\kappa^2$-O$_2$N)$_2$(L1″)], Figure S14: EPR spectra of [Fe(NO)$_2$(L1″)](BF$_4$) at 130 K, Figure S15: $^1$H-NMR spectrum of [FeCl$_2$(L1″)], Figure S16: $^1$H-NMR spectrum of [Co(NO$_3$)$_2$(L1″)], Figure S17: $^1$H-NMR spectrum of [Co(NO$_2$)$_2$(L1″)], Figure S18: IR spectral changes under argon atmospheres at room temperature (KBr), Figure S19: IR spectral changes during the O$_2$ reaction of [Fe(NO)$_2$(L1″)](BF$_4$) in solution (CH$_2$Br$_2$), Figure S20: IR spectral changes during the O$_2$ reaction of [Fe(NO)$_2$(L1″)](BF$_4$) in the solid (CH$_2$Br$_2$), Figure S21: IR spectral changes during the O$_2$ reaction of [Co(NO)$_2$(L1″)](BF$_4$) in solution (CH$_2$Br$_2$), Figure S22: IR spectral changes during the O$_2$ reaction of [Co(NO)$_2$(L1″)](BF$_4$) in the solid (CH$_2$Br$_2$), Figure S23: $^1$H-NMR spectrum of the products of the O$_2$ reaction of [Co(NO)$_2$(L1″)](BF$_4$) in solution, Figure S24: $^1$H-NMR spectrum of the products of the O$_2$ reaction of [Co(NO)$_2$(L1″)](BF$_4$) in the solid.

**Author Contributions:** K.F. conceived and designed the project. H.K. and A.O. performed the experiments. H.K. and K.F. analyzed the data. K.F. wrote the paper.

**Funding:** This research received no external funding.

**Acknowledgments:** We are grateful for the support from the Japan Society for the Promotion of Science (JSPS) and an Ibaraki University Priority Research Grant.

**Conflicts of Interest:** The authors declare no conflict of interest.

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
