# Peer review of "Structures and Properties of Dinitrosyl Iron and Cobalt Complexes Ligated by Bis(3,5-diisopropyl-1-pyrazolyl)methane"

_inorganics, doi:10.3390/inorganics7100116_

Round 1

Reviewer 1 Report

This paper on M(bidentate)2(NO)2 complexes, M = Fe, Co) follows on from related work by the authors and is generally of a reasonably high standard.

p.5, p. 12   omit Prof from Prof Lippard and Prof Kim

p. 8 line 202   IR using KBr pellets   could M(L)2(Br)2 be forming?

IR section- I recommend shortening considerably

Uv-visible spectra - do the authors see any d-d- bands for these tetrahedral species?

p. 15 line 398   isotopic  not isotropic

O2 reactivity   esr spectra for Co systems would be helpful   in case any O2(-) intermediates are formed. The O2 work is inconclusive and should be shortened

In the Fe and Co cases, magnetic moments (vs temperature)  and Mossbauer spectra (Fe) would help identify metal oxidation and spin states; theoretical calculations (e.g. DFT) would improve quantitative aspects of the work. For instance, diamagnetic tetrahedral Co(II??) is uncommon.

Throughout the script, English grammar needs checking; many "the" are missing

In summary; the paper is just acceptable   and would be improved by the suggestions given above.

Author Response

This paper on M(bidentate)2(NO)2 complexes, M = Fe, Co) follows on from related work by the authors and is generally of a reasonably high standard.

Thank you very much for your supports for our publication.

p.5, p. 12   omit Prof from Prof Lippard and Prof Kim

We omitted “Prof.” for the main text as you suggested.

p. 8, line 202   IR using KBr pellets   could M(L)2(Br)2 be forming?

We did not observe this bromination. If this bromination would occur, we can observe the color changes.

IR section- I recommend shortening considerably

Now we shorten the section 2.3 (IR) and the section 2.6 (reactivity). We moved 3 figures (Figures 6-8) to the SI. The section 2.6 is now 1.5 pages from 3 pages. This IR section (section 2.3) is very important in this manuscript, since all of DNICs were discussed by ν(N-O). And as mentioned in the text, IR stretching values can conclude their electronic structures. Therefore, these discussions including 15NO labeling experiments are very important. Moreover, it is very few examples to show ν(M-N(O)). From your suggestions, we deleted the IR discussions of the related Co(II) complexes.

Uv-visible spectra - do the authors see any d-d- bands for these tetrahedral species?

We tried to say the coordination geometry from Figure 4 (Figure 4 was moved to SI.) However, it is very weak in intensity, so this geometry discussion was deleted in the main text. Now, section 2.4 is only 0.5 page.

p. 15 line 398 isotopic not isotropic

Thank you very much for your comments. We tried to reduce these errors over the text.

O2 reactivity esr spectra for Co systems would be helpful in case any O2(-) intermediates are formed. The O2 work is inconclusive and should be shortened

In the Fe and Co cases, magnetic moments (vs temperature)  and Mossbauer spectra (Fe) would help identify metal oxidation and spin states; theoretical calculations (e.g. DFT) would improve quantitative aspects of the work. For instance, diamagnetic tetrahedral Co(II??) is uncommon.

Thank you very much for your suggestions. This Co reactivity is very slow; therefore, we cannot observe any intermediates in UV-Vis spectra. We are now underway to trap these intermediates by both Fe and Co dinitrosyl complexes. We will note the results in the near future. We did not have any variable temperature magnetic moment and Mössbauer spectroscopy. We need more time to use these special techniques. DFT is very helpful, however, we are also underway to know the metal oxidation states and spin state with collaborations. Therefore, it is also the next step. For this ambiguity, we used “Enemark−Feltham notation”. I know some of DINICs show only this notation as the conclusions. Therefore, our conclusion of {Fe(NO)2}9 and {Co(NO)2}10 is not so special one.

Throughout the script, English grammar needs checking; many "the" are missing

Thank you very much for your comments. We tried to reduce these errors over the text.

In summary; the paper is just acceptable and would be improved by the suggestions given above.

Thank you very much for your positive comments for our submission. We wish this revised manuscript is enough for the publication.

Reviewer 2 Report

This manuscript by Fujisawa and co-workers presents the results of their investigations surrounding iron and cobalt nitrosyl complexes. There appears to a good body of results within the manuscript however I believe that the work is poorly presented and I would not recommend publication in this form. The presentation of the work is difficult for the reader to follow and as such it is challenging for the reviewer to assess the merits of this work. I would suggest that the authors consider rewriting this manuscript and resubmitting.

Some of the main points which I suggest improvements are needed are:

1. The title needs changing N2 looks like dinitrogen no bis-nitrogen donor ligands.

2. The abstract should be written in the third person

3. The ligand L'' is not defined in the abstract. It does not make sense to define the only ligand used as L''

4. The abstract and introduction sections need more of a clear narrative. It is not really clear what the authors are trying to present. Why have these specific complexes been targeted? The way in these have been presented seem to suggest that the complexes were randomly chosen.

5. Again, why start with a ligand defined as L3? How are the two ligands related? Why is this comparison necessary?

6. Dinitrosyl is spelt incorrectly a number of times within the manuscript.

7. The synthesis of the complexes is not outlined clearly enough.

8. Line 116 - "N11-M-N21" not "N11-M-N2"

9. Line  135 - "solvents of crystallization" not "crystalline solvents"   

10. delete 2 x "cation portion of...) in Figure 1. It already says BF4 counterion omitted.

11. "We try to assign...." needs to be changed

12. Why is some of the main text in bold on page 9?

13. Figures 6-8 are not the best spectra and could be improved

With some significant improvements in the presentation, I believe that this work would be of interest to the readers of Inorganics.  

Author Response

This manuscript by Fujisawa and co-workers presents the results of their investigations surrounding iron and cobalt nitrosyl complexes. There appears to a good body of results within the manuscript however I believe that the work is poorly presented and I would not recommend publication in this form. The presentation of the work is difficult for the reader to follow and as such it is challenging for the reviewer to assess the merits of this work. I would suggest that the authors consider rewriting this manuscript and resubmitting.

Thank you very much for your following comments. We rewrite the main text accordingly. So, we wish that the current version is enough for your science standards.

Some of the main points which I suggest improvements are needed are:

The title needs changing N2 looks like dinitrogen no bis-nitrogen donor ligands. We changed the title as your suggestion to “Ligated by Bis(3,5-diisopropyl-1-pyrazolyl)methane”.

The abstract should be written in the third person

Yes, I wrote this abstract.

The ligand L'' is not defined in the abstract. It does not make sense to define the only ligand used as L''

Thank you very much for your comments. I inserted the abbreviations in the Abstract.

The abstract and introduction sections need more of a clear narrative. It is not really clear what the authors are trying to present. Why have these specific complexes been targeted? The way in these have been presented seem to suggest that the complexes were randomly chosen.

I tried to rewrite both Abstract and Introduction parts which were marked as the yellow makers. Lines 50-59, I tried to note our motivations in this research. I wish you can understand this current revised version.

Again, why start with a ligand defined as L3? How are the two ligands related? Why is this comparison necessary?

This is also noted in lines 50-59, as mentioned in No. 4 point.

Dinitrosyl is spelt incorrectly a number of times within the manuscript.

Thank you very much for your note. I checked them over the text.

The synthesis of the complexes is not outlined clearly enough.

Thank your very much for your opinion. I changed some sentence which were marked as the yellow makers.

Line 116 - "N11-M-N21" not "N11-M-N2"

I changed them, thank you very much for our careless mistakes. We changed these mistakes for over the text.

Line 135 - "solvents of crystallization" not "crystalline solvents"

I changed these words over the text, thank you very much.

delete 2 x "cation portion of...) in Figure 1. It already says BF4 counterion omitted.

I deleted them as your suggestions.

"We try to assign...." needs to be changed

I deleted these words and rewrote them.

Why is some of the main text in bold on page 9?

I am not sure, however, I changed them.

Figures 6-8 are not the best spectra and could be improved

These figures were moved to SI. And this section (reactivity) was condensed to 1.5 pages from 3 pages.

With some significant improvements in the presentation, I believe that this work would be of interest to the readers of Inorganics.

Thank you very much for your positive comments for our submission. We wish this revised manuscript is enough for the publication.

Reviewer 3 Report

In this paper the synthesis and characterization of two dinitrosyl iron and cobalt complexes is reported. All the characterizations seem to have been made sufficiently well and the paper could be considered for publication in Inorganics. Only minor points:

1) The contents of tables 3 and 4 are different from what declared (... iron (3) or cobalt (4) complexes).

2) It is strange, from my point of view, that the magnetic properties have been investigated only by NMR measurements.

Author Response

In this paper the synthesis and characterization of two dinitrosyl iron and cobalt complexes is reported. All the characterizations seem to have been made sufficiently well and the paper could be considered for publication in Inorganics.

Thank you very much for your positive comments for our submission. We wish this revised manuscript is enough for the publication.

Only minor points:

The contents of tables 3 and 4 are different from what declared (... iron (3) or cobalt (4) complexes).

I am sorry for this point. I rewrote them in lines 615-617 to easily understand.

It is strange, from my point of view, that the magnetic properties have been investigated only by NMR measurements.

That is right. During this revised version, we tried to access EPR spectroscopy. We added the preliminary EPR spectrum in Figure S14 and the total spin state (St = 1/2) discussion was added in the main text.

Reviewer 4 Report

The manuscript entitled “Structures and Properties of Dinitrosyl Iron and Cobalt Complexes by Using N2 Type Ligand” submitted by Kisyoshi Fujisawa et al. describes the syntheses of two dinitrosyl iron and cobalt complexes. Though the reactivity of these complexes toward dioxygen is not clear, the syntheses is interesting. Based on the novelty of syntheses, the reviewer recommends the acceptance for publication in Inorganics.

Author Response

The manuscript entitled “Structures and Properties of Dinitrosyl Iron and Cobalt Complexes by Using N2 Type Ligand” submitted by Kisyoshi Fujisawa et al. describes the syntheses of two dinitrosyl iron and cobalt complexes. Though the reactivity of these complexes toward dioxygen is not clear, the syntheses is interesting. Based on the novelty of syntheses, the reviewer recommends the acceptance for publication in Inorganics.

Thank you very much for your positive comments for our submission. We wish this revised manuscript is enough for the publication.

Round 2

Reviewer 2 Report

The authors have made a number of improvements to the manuscript and on this basis I would recommend that it is accepted subject to the following further minor changes.

There are two ligands presented in the manuscript. For clarity of presentation these should be given consecutive numbering. I would call them L1 and L2. (Apologies, the typo in my original report might have confused my previous recommendation). Any additional ligands referred to should be labelled L3, L4 and so on... The abstract should be presented in the third person (i.e. "Two dinitrosyl iron and cobalt complexes, [Fe(NO)2(L1”)](BF4) and [Co(NO)2(L1”)](BF4) were synthesized.....). The authors should avoid using "we" so much within the manuscript. 

Author Response

[Reviewer 2]

The authors have made a number of improvements to the manuscript and on this basis I would recommend that it is accepted subject to the following further minor changes.

Thank you very much for your positive comments for our submission. We wish this revised manuscript is enough for the publication.

There are two ligands presented in the manuscript. For clarity of presentation these should be given consecutive numbering. I would call them L1 and L2. (Apologies, the typo in my original report might have confused my previous recommendation). Any additional ligands referred to should be labelled L3, L4 and so on...

Thank you very much for your comments. These comments were sometimes found from some reviewees. However, these nomenclatures were very famous by our publications. Hydrotris(3-tertiary butyl-5-isopropyl-1-pyrazolyl)borate is always called as “L3” in the recent our publications, see Inorg. Chem. 2019, 58(7), 4059–4062, Dalton Transductions, 2017, 46(39), 13273–13289, J. Biol. Inorg. Chem. 2017, 22, 237–251, J. Biol. Inorg. Chem. 2016, 21, 757–775, Coord. Chem. Rev. 2013, 257(1), 119–129. Therefore, Cu2O2 chemist such as Prof. Ed Solomon, Prof. Ken Karlin, and Prof. Dan Stack used this nomenclature. In special, Prof. Ed Solomon used this L3 in his presentation. Therefore, I would like to keep these systematic nomenclatures. Our ligand system: L0 3,5-Me2-1-pzH, L1 3,5-iPr2-1-pzH, L2 3-tBu-5-Me-1-pzH, L3 3-tBu-5-iPr-1-pzH, L4 3,5-tBu2pzH.And L means hydrotris(pyrazolyl)borate, L' means tris(pyrazolyl)methane, L1'' means bis (pyrazolyl)methane.

The abstract should be presented in the third person (i.e. "Two dinitrosyl iron and cobalt complexes, [Fe(NO)2(L1”)](BF4) and [Co(NO)2(L1”)](BF4) were synthesized.....). The authors should avoid using "we" so much within the manuscript.

Thank you very much for your point. I changed this sentence accordingly. I am sorry not to understand your comments in the 1st revision.
